# Moderating Effect of Changes in Perceived Social Support during Pregnancy on the Emotional Health of Mothers and Fathers and on Baby’s Anthropometric Parameters at Birth

**DOI:** 10.3390/children9050648

**Published:** 2022-04-30

**Authors:** María José Castelar-Ríos, Macarena De los Santos-Roig, Humbelina Robles-Ortega, Miguel Ángel Díaz-López, José Maldonado-Lozano, Mercedes Bellido-González

**Affiliations:** 1Faculty of Psychology, University of Granada, 18071 Granada, Spain; mjcastelarrios@gmail.com (M.J.C.-R.); dlsantos@ugr.es (M.D.l.S.-R.); 2Gynecology Service, Virgen de las Nieves University Hospital, University of Granada, 18071 Granada, Spain; mdiazl@ugr.es; 3Pediatric Service, Virgen de las Nieves University Hospital, University of Granada, 18071 Granada, Spain; jmaldon@ugr.es; 4Faculty of Education Sciences, University of Granada, 18071 Granada, Spain; mmbellid@ugr.es

**Keywords:** depression, stress, perceived social support, protective factors, birth weight, cephalisation index, mother, father, baby

## Abstract

(1) Background: this study is based on a model of how changes in protective factors may affect the emotional health of mothers and fathers and thus influence the development of the baby. Our research goal is to determine whether variations in perceived social support moderate levels of stress and depression during pregnancy and/or the effect of parents’ emotional health on the baby’s anthropometric parameters. (2) Methods: to achieve these aims, a longitudinal study was made of 132 couples and babies, who were evaluated at weeks 12 and 32 of gestation and at birth. Separate analyses were performed for the mothers and fathers, focused on the role of social support in moderating their levels of depression and stress during pregnancy, and the consequent impact on the baby. (3) Results: the results obtained show the moderating effects of changes in social support on maternal and paternal stress and depression. Reduced social support during pregnancy is associated with higher levels of stress and depression in both parents and with a high cephalisation index in their babies. (4) Conclusions: special attention should be paid to social support, which can have a strong impact on the evolution of emotional health during pregnancy and concomitantly on the development of the baby.

## 1. Introduction

Pregnancy is known to be associated with intense emotional changes [1,2]. Studies have revealed the existence of distinctive emotional patterns in pregnant women, who often present non-clinical symptoms of depression and/or stress. These reactions are more acute during the first and third trimesters, and somewhat less so during the second [2,3]. The latter authors remarked that this pattern might be related, at least in part, to heightened concerns about the evolution of the pregnancy during the first trimester (such as fear of miscarriage or the discomfort caused by nausea and/or vomiting, fatigue, hormonal changes, etc.), and during the final stages (concerns about mobility, physical changes, the proximity of the birth, etc.). Although these changes are commonly experienced, they make future mothers more vulnerable to major depressive disorder and stress [4]. In addition, these outcomes are closely related, in the sense that stress can contribute to the appearance of depressive problems [5]. Accordingly, stress and depression should be jointly analysed within a single model of emotional health in pregnant women.

It is of crucial importance to understand the consequences of this emotional pattern, not only for the interests of the expectant mother, but also because stress and depression may have a predictive capacity for the anthropometric parameters of the baby at birth [6,7]. Exposure to stress can produce a dysregulation of the hypothalamic–pituitary–adrenal (HPA) axis, and provoke cortisol release in pregnant women. This mechanism jeopardises anthropometric foetal development and can result in adverse birth outcomes [8,9].

However, protective factors such as social support can prevent or cushion both depression and maternal stress. Perceived social support is the understanding that psychosocial help is available when needed. This type of support produces tangible benefits during pregnancy and can be reliably measured during this period [10]. Studies have reported that the existence of perceived social support during pregnancy limits the development of emotional problems in the mother [11,12]. This support can be obtained from various sources, but it seems that only family and partner support is really effective in preventing or buffering depression and stress [11]. In addition, social support may change during the course of pregnancy, thus producing a differentiating effect on parental stress and depression.

Consequently, the model on which our study is based is evolutionary and conceptualised in terms of change. Assuming that psychological changes during pregnancy can act as stressors, their effects may be differentially moderated by changes in social support. Accordingly, the resulting stress and depression will be determined, at least in part, by the evolution of social support (phase 1, Figure 1), which can also influence the baby’s anthropometric parameters (phase 2, Figure 1) [13,14].

In line with this model, the hypothesis considered in this study is that a change in social support during pregnancy will provoke a corresponding change in levels of maternal and paternal stress and depression.

Another consideration is that variations in social support for the parents might influence the infant’s clinical outcome. At present, it remains unclear whether the moderating factors of stress and depression have any such impact [15].

Although many studies have analysed the situation of the expectant mother, the emotional health of the father during pregnancy has received far less research attention [16]. Nevertheless, men are also subject to emotional problems during this period [17], which can provoke alterations in the baby’s subsequent behaviour and neurodevelopment [18].

Therefore, as well as applying the model of analysis to the mothers in the study sample, we believe it useful to consider the effects of changes in the social support perceived by the fathers, with respect to their emotional health and to the condition of the infant at birth [19].

In view of these considerations, our research has two main objectives: (1) to examine whether changes in social support (positive or negative) have protective or prejudicial effects during pregnancy; that is, whether such changes moderate the stress and/or depression experienced during pregnancy; (2) to determine whether social support during the third trimester also moderates the effect of the parent’s emotional state on the baby’s anthropometric parameters.

## 2. Methods

### 2.1. Participants

For the purposes of this prospective longitudinal study, 132 women in the first trimester of pregnancy and their partners, 132 expecting fathers, were recruited, from the 1555 women who attended the level-three Virgen de las Nieves Hospital in Granada (Spain) during the six-month period from October 2017 to March 2018. Table 1 summarises the characteristics of these participants at 12 weeks gestation.

The following criteria were applied for inclusion in the study: both the pregnant woman, in the twelfth week of pregnancy, and her partner should provide signed informed consent to take part. The exclusion criteria were: (1) currently receiving psychiatric treatment; (2) drug use; (3) foetal death; (4) failure to attend face-to-face assessments for work-related reasons.

The mothers and their partners were evaluated at two time points during the pregnancy: at 12 weeks (GT-12) and at 32 weeks (GT-32). After applying the exclusion criteria, the sample at GT-12 consisted of 130 mothers and 106 expecting fathers (see flowchart in Figure 2).

The longitudinal follow-up of stress and depression was completed by 96.96% of the mothers in our sample, but only by 79% of the fathers. This difference arose because the evaluations took place in person at the hospital, and the consultations were not always attended by both parents. The male partner attended less often, usually because of work-related obligations. No differences were found in the sociodemographic variables between those who dropped out and those who continued the study, either for mothers or fathers.

### 2.2. Measures

Edinburgh Postnatal Depression Scale (EPDS) [20,21]. This 10-item scale assesses the subject’s mood during the previous week (ranging from 0 “always or most of the time” to 3 “never”). The Spanish version of EPDS provides good psychometric properties [20]. In our sample, Cronbach’s α was 0.72.

Perceived Stress Scale (PSS) [22,23]. This instrument consists of 14 items related to life situations that may have been stressful in the previous month (range of response from 0 “never” to 4 “very often”). The scale presents acceptable reliability (α = 0.81 and R test-retest = 0.73) and good validity and sensitivity [22]. In our sample, Cronbach’s α in this respect was 0.80.

The Multidimensional Scale of Perceived Social Support (MSPSS) [24]. This 12-item scale assesses the social support received. Possible responses range from 1 “strongly disagree” to 5 “strongly agree”. The MSPSS provides good reliability, with a Cronbach’s α of 0.90 [25]. In our sample, α = 0.89.

Gynaecological Assessment of Pregnancy. The gynaecological follow-up procedure was carried out according to the protocol described in NICE (2008) [26].

Anthropometric measurements of the newborn were conducted in the delivery room by clinical staff using standard protocols and instrumentation. A Seca electronic baby scale (Seca Ltd., Hamburg, Germany) was used to measure infant weight to the nearest 1 g. A Seca infantometer was used to measure supine length to the nearest 0.1 cm. To measure head circumference, a Seca 212 measuring tape was circled around the largest circumference of the head. Weight and head circumference were used to calculate the cephalisation index (head circumference cm/birthweight g) × 100). This index reflects the degree of asymmetric intrauterine growth, with higher scores indicating potential brain sparing during gestation and offspring neurodevelopmental vulnerability [15,27,28].

### 2.3. Procedure

In this study, all procedures involving human participants were performed in accordance with the ethical standards of the institutional and/or national research committee and with the 1964 Helsinki Declaration and its later amendments, or comparable ethical standards. The protocol was approved by Andalusian Biomedical Research Ethics Committee (Spain) (Project identification code: PC-0526-2016-0526) and approved 30 November 2016.

All participants in this study were informed by gynaecologists about the advisability of assessing their emotional health in each trimester during the pregnancy, in parallel with the gynaecological monitoring programme.

Couples whose pregnancy was treated at the HUVN during the last quarter of 2017 and the first quarter of 2018 received the corresponding gynaecological attention, in accordance with the NICE guidelines (2008) [26].

The couples who agreed to participate in the study were referred to collaborating psychologists, who conducted extensive assessments with the mother and her partner. At the first meeting, their informed consent to participate was requested and obtained, and information was sought about their sociodemographic characteristics (age, level of education and employment status).

Self-report questionnaires were distributed and completed to facilitate an assessment of the participants’ emotional health during the first trimester of pregnancy. These questionnaires were focused on identifying perceived stress and/or depression and social support, and were presented in a counterbalanced order. The assessment was conducted in a quiet, secluded room in the hospital, offering adequate privacy. The same instructions were given in all cases.

The same procedure was followed in the third trimester of the pregnancy (GT-32), during which the evaluation of the couples’ emotional health was coordinated with the gynaecological monitoring in order to minimise the number of hospital visits and thus ensure, as far as possible, continuity in the longitudinal analysis.

### 2.4. Data Analysis

A series of moderation analyses were performed to analyse the role of perceived change in social support in moderating differences in depression (S12 with respect to S32) and in stress (S12 with respect to S32), during pregnancy (phase 1, Figure 1). We then considered the moderating role of social support at GT-32 with respect to alleviating any adverse relationship between the parents’ depression/stress and the babies’ anthropometric measures at birth (phase 2, Figure 1).

Social support is a continuous variable and so various hierarchical multiple regressions were performed after controlling for covariates. These covariates were selected a priori based on their potential to confound the associations [29] and included demographic (age, employment and educational attainment), maternal health (health risks and number of pregnancies), delivery (type), and infant-related factors (gestational age).

Moderation hypotheses appear in every area of psychological science, but the methods for testing and probing moderation in two-instance repeated-measures designs were less developed. To accomplish that, the MEMORE (Mediation and Moderation in Repeated-Measures Designs) macro for SPSS was developed by A.K. Montoya [30]. MEMORE was used for the analysis of the phase 1 results. The two gestational moments (GT-12 and GT-32) were predictors of differences in depression and stress, over time, and change in social support was a moderating factor. The results for phase 2 were analysed using the PROCESS macro for SPSS developed by Hayes (2013) [31]. PROCESS is an observed variable ordinary least squares and logistic regression path analysis modelling tool. It is widely used in the social and health sciences for estimating direct and indirect effects in single and multiple mediator models, but also in moderation models. In this work, PROCESS was used in order to test the moderating role of social support at GT-32 on the relation between parents’ depression/stress and babies’ anthropometric measures.

## 3. Results

Before analysing the result for phase 1, the equivalence between the fathers and mothers with respect to age, level of education and employment status (working vs. non-working) was considered. Table 1 shows there were differences in age (*p* < 0.01) and in the proportions of those in employment (*p* < 0.01). However, levels of education were similar for men and women.

### 3.1. Aim 1: To Determine the Outcomes of Perceived Changes in Social Support during Pregnancy, with Particular Reference to the Moderation of Depression and Stress in This Period

Social support was significantly related to depression and stress at both timepoints considered. For mothers, lower perceived social support at GT-12 predicts higher levels of depression/stress, and vice versa, at this moment of pregnancy (for depression, R^2^ = 0.098, F = 13.60, *p* < 0.001; β_GT-12_ = −0.313, t = −3.68, *p* < 0.001; and for stress, R^2^ = 0.131, F = 19.11, *p* < 0.001; β_GT-12_ = −0.362, t = −4.37, *p* < 0.001). The same relation was observed at the end of pregnancy (for depression, R^2^ = 0.196, F = 30.62, *p* < 0.001; β_GT-12_ = −0.442, t = −5.53, *p* < 0.001; and for stress, R^2^ = 0.151, F = 23.18, *p* < 0.001; β_GT-12_ = −0.389, t = −4.81, *p* < 0.001). Among the fathers, similar results were obtained at the beginning of pregnancy (for depression, R^2^ = 0.102, F = 14.42, *p* < 0.001; β_GT-12_ = −0.193, t = −3.76, *p* < 0.001; and for stress, R^2^ = 0.186, F = 28.82, *p* < 0.001; β_GT-12_ = −0.432, t = −5.36, *p* < 0.001) and at the end (for depression, R^2^ = 0.129, F = 14.71, *p* < 0.001; β_GT-12_ = −0.360, t = −3.83, *p* < 0.001; and for stress, R^2^ = 0.134, F = 15.78, *p* < 0.001; β_GT-12_ = −0.366, t = −3.97, *p* < 0.001). Thus, lower social support was associated with higher scores for depression/stress and vice versa.

The effects of changes in perceived social support during pregnancy (positive or negative) and the question of how depression and stress might be affected by these changes were examined as follows. For each of the two time points (at the beginning and at the end of the pregnancy), changes in perceived social support were considered as moderators of differences in depression and/or stress during pregnancy, for the mothers and the fathers separately. Following these analyses, groups of improved, impaired and no change social support were generated following the 16th, the 50th and the 84th percentiles of the differences between GT-12 and GT-32 in social support. No differences were found between those groups regarding age, employment status or education levels, from which we conclude that none of these variables accounts for changes in depression or stress.

Analysis showed that a change in perceived social support during pregnancy was a significant moderator of differences in depression and of stress in this period, accounting for about 15% (depression) and 11% (stress) of the variance in the mothers and 8–10% of the variance in the fathers (see Table 2).

The repeated measures *t*-test (see Table 3) showed that among the mothers whose perceived social support increased during pregnancy (*n* = 23), the scores for depression fell significantly (*p* < 0.001). In contrast, when perceived social support fell (*n* = 24), the opposite effect on depression was observed (*p* < 0.01). Similar results were obtained for the fathers who perceived increased social support (*n* = 17). The resulting fall in depression was statistically significant (*p* < 0.01). Among those who perceived lower levels of social support (*n* = 18), the levels of depression rose significantly (*p* = 0.05). These findings, together with those for the ‘no change in social support’ group, are detailed in Table 3.

Regarding the moderating effect on stress of changes in social support, the results obtained reflect a similar pattern to that observed for depression, both for mothers and for fathers (Table 3). In the mothers who perceived improved social support (*n* = 23), levels of stress fell significantly (*p* = 0.05). In contrast, when the social support worsened, significantly higher levels of stress were experienced (*p* < 0.001). Among the fathers, when perceived social support rose (*n* = 18), levels of stress diminished significantly (*p* < 0.001). When this support fell, levels of stress rose but the change was only marginally significant (*p* < 0.08). These findings, together with those for the ‘no change in social support’ group, are detailed in Table 3.

### 3.2. Aim 2: Social Support as a Mitigating or Protective Factor Moderating the Relationship between Parents’ Levels of Depression/Stress and Their Babies’ Anthropometric Measures

In addition to the above, we examined whether perceived social support at GT-32 moderated the effects of parents’ depression and stress on their babies’ cephalisation index and size at birth. These analyses were performed, after controlling for covariates, for the mothers and fathers separately.

Table 4 shows the contribution of each factor, both individually (the isolated effects of depression/stress and of social support) and in interaction (interaction effects) on the babies’ anthropometric measures. These results show that maternal depression and stress at GT-32 are strongly related to the cephalisation index at birth; thus, the greater the depression and/or stress experienced, the higher the cephalisation index (in both cases, *p* < 0.01). Social support alone has a marginally significant impact on depression/stress.

However, the interaction between the two factors was statistically significant. Thus, the “Social support × Depression” and the “Social support × Stress” interactions enabled us to identify the following groups: low/high social support and low/high depression/stress. Subsequent *t*-test analysis revealed that the interaction effect was significant for the low social support group at GT-32. The clinical results for the babies varied significantly according to the scores recorded for maternal depression/stress. High scores were associated with a higher cephalisation index (mean values = 1.18 vs. 1.06, t = 3.13, *p* = 0.02; mean values = 1.18 vs. 1.05, t = 2.94, *p* = 0.004, respectively, with high/low depression or high/low stress). For the remaining groups derived from the interaction (high social support and high/low depression/stress or any other combination), the results were not statistically significant.

With respect to the results for the babies’ size at birth, the results were only marginally significant in almost all cases (see Table 4). Finally, the fathers’ scores for social support or depression/stress at GT-32 bore no direct relation (either as a single effect or as an interaction) with any of the anthropometric measures.

## 4. Discussion

The main aims of this study were to determine whether changes in social support influence depression and/or stress in mothers and fathers during pregnancy and whether the moderating role of this support during the third trimester of pregnancy influences the anthropometric parameters of babies at birth.

In line with previous research findings, our results show that in the first and third trimesters of pregnancy, levels of stress and depression in the parents are moderated by their perceived social support; thus, higher levels of support are associated with lower levels of depression and stress, and vice versa [32,33]. Many other studies have analysed the evolution of maternal health during pregnancy, but to our knowledge, none have considered the moderating effect on these factors of perceived social support at different time points.

The first major contribution of the present study is the confirmation that the presence and evolution of maternal and paternal stress and depression are influenced by perceived social support, and that changes in this moderator variable are associated with corresponding changes in stress and depression. Thus, the parents whose perceived social support was low in the first trimester of the pregnancy but higher in the third subsequently presented lower levels of stress and depression. In other words, the improvement in perceived support alleviated the stress and depression experienced. By contrast, when perceived social support worsened, levels of depression and stress rose towards the end of the pregnancy. When social support remained unchanged, so did stress and depression.

The same pattern was observed regarding the moderating impact of perceived social support on paternal stress and depression during the pregnancy. Unlike other studies of the relationship between social support and stress/depression at different stages of pregnancy [11] which have focused only on the effects produced on the mother, we also examined this relationship for the fathers. The results obtained show that social support is a moderating factor and that changes in this variable produce corresponding changes in levels of paternal stress/depression. In this respect, hence, similar reactions are experienced by both parents.

To date, most studies of emotional health during pregnancy have focused on the mother. Indeed, there is still a certain resistance to the inclusion of the father in such studies, despite the significance of the partner in the relationship. According to the few studies that have included fathers in their analysis, it is clear that both parties experience stress and depression during pregnancy, although the prevalence and evolution of these problems during this time remains an open question. Indeed, many men believe that insufficient account is taken of their own emotional health during the pregnancy and that support and resources should be provided in this respect [18]. In view of these considerations, our study seeks to determine the role played by social support in protecting the emotional health not only of the mother—as reported by other authors [11,12]—but also of the father and the baby. Our findings confirm those of previous research and extend them to show that this factor also protects fathers against stress and depression during pregnancy.

The final notable contribution of this study is to determine the relationship between social support and the anthropometric parameters of the baby at birth, showing that perceived social support is a moderator variable of stress and depression in both parents. Like previous studies, we found that those mothers who perceived less social support at GT-32 had higher levels of stress and depression. Moreover, their babies presented a higher cephalisation index at the time of birth [13,15,34], and hence an aggravated neonatal status. This effect was also observed for size at birth, albeit less strongly. In addition, and unlike previous studies, we also tested whether the social support received by fathers during the third trimester of pregnancy had any influence on the birth outcomes of the babies. The results obtained revealed no such effect. Ultimately, the anthropometric parameters of babies are directly determined by the activity of the mother’s HPA axis and cortisol burden [9], which are related to levels of stress, depression and perceived social support. So, the expectant woman’s perception of emotional discomfort in the father does not trigger the same physiological and psychological responses in herself, or with the same intensity, as undergoing this directly. Additionally, it is probably for this reason that no strong interaction was observed between the emotional health of fathers and the anthropometric parameters of the baby at birth at GT-32 in our study.

In the field of study considered, it is already known that the emotional health of pregnant women has important consequences on birth outcomes and the babies’ subsequent neurodevelopment [7,35,36]. Extending our understanding, the present study shows, moreover, that social support for pregnant women is a protective variable for the neurodevelopment of their babies, with beneficial effects that can become apparent in the short term.

These findings have important implications for the design of interventions aimed at supporting emotional health during pregnancy and improving babies’ neurodevelopment. Such interventions are especially indicated during the first and third trimesters, when stress and depression present the greatest variability and are most susceptible to the influence of social support. Multidisciplinary teams, with specialists in gynaecology and psychology, should be created so that a comprehensive approach can be taken to assist families during pregnancy, thus preventing or alleviating problems of physical and emotional health that might prejudice foetal and postnatal development [7].

To date, most interventions in this field have addressed approaches to stress and depression such as cognitive behavioural therapy, behavioural activation and/or mindfulness programmes [37]. However, in view of the findings obtained in the present study, attention should also be paid to fostering social support, both for the mother and for the father, in order to achieve optimal outcomes for their babies. Further information is needed about the quantity and type of support that would be most effective in alleviating stress and depression, and about the type of psychosocial interventions which should be employed during pregnancy.

## 5. Limitations

The present study has certain limitations that should be taken into account. First, fewer fathers than mothers took part in the whole study, probably because employment obligations prevented them from regularly attending appointments. Nevertheless, the majority of those who did participate were present at both sessions. Second, the number of male participants lost to the study might have reduced the representativeness of this sample. However, the analyses carried out of the socio-demographic variables revealed no significant differences in this respect.

Offsetting these weaknesses, our study has several strengths. First, only a prospective, longitudinal study such as the one we describe shows how the parameters in question evolve over time. Second, the psychologists who participated were also experts in the evaluation of psychological disorders, and this quality reduced the possibility of reporting bias. Third, the questionnaires were completed in person at the hospital, with the same conditions for all participants. Finally, maternal and paternal stress, depression and perceived social support were evaluated by repeated measures, which reduced the possibility of errors in the classification of results.

## 6. Conclusions

The moderating effect of social support is powerful, such that a change in this protective variable produces changes in stress and depression outcomes, for both parents.Babies whose mothers were more depressed or stressed presented higher rates of cephalisation.Interventions tailored to each stage of pregnancy should be implemented to provide parents with the means to reinforce social support as a protective factor.Finally, the present study is focused on parents whose pregnancy follows the normal course. However, it may be assumed that if an intervention to foster social support is beneficial for these mothers and fathers, it would be even more so for those having to cope with a high-risk pregnancy.

## Figures and Tables

**Figure 1 children-09-00648-f001:**
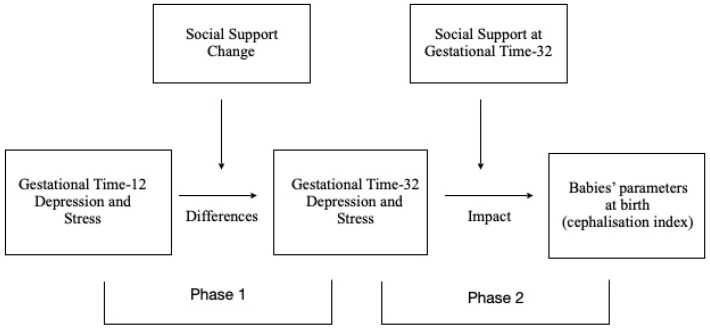
Evolutionary model tested, with two phases: (1) the moderating role of social support change on differences in depression and stress during pregnancy; (2) the moderating role of social support at GT-32 on the impact of parents’ depression and stress on babies’ anthropometric measures at birth.

**Figure 2 children-09-00648-f002:**
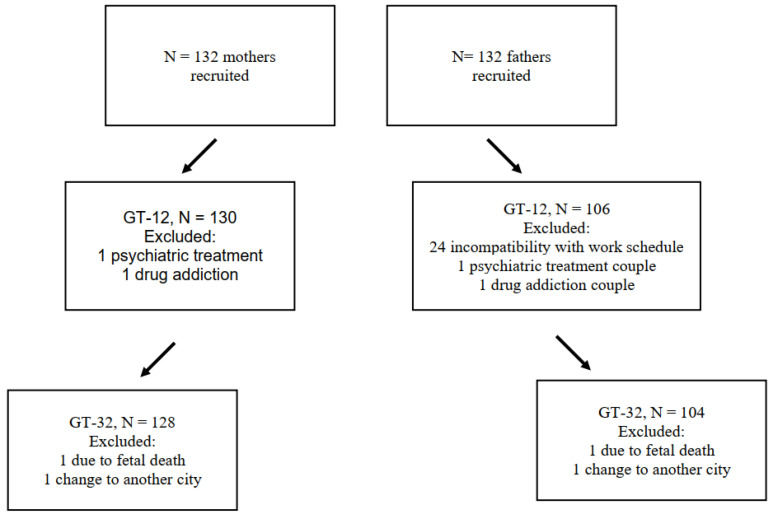
Flowchart of mothers and fathers at the time points evaluated (GestationTime-12, GestationTime-32).

**Table 1 children-09-00648-t001:** Characteristics of the pregnant women and their partners at the start of the study, week 12 of pregnancy (N = 130 mothers, N = 106 partners).

	Mother (N = 130)	Partner (N = 106)	Test
Age, mean (SD)	30.53 (5.79)	32.87 (5.69)	t_222_ = −3.80 **
Range	16–44	16–47	
Duration of relationship, mean (SD)	7.58 (6.53)	7.58 (6.53)	
Range	1–25	1–25	
Parity status, *n* (%)			
Primiparous	69 (53.37)		
Multiparous	61 (46.63)		
Spanish language fluency, *n* (%)	130 (100)	106 (100)	
Education, *n* (%)			
Primary	39 (30.06)	32 (29.83)	
Secondary	44 (33.74)	35 (33.33)	χ^2^ = 5.00 n.s.
Higher	47 (36.20)	39 (36.84)	
In paid employment, *n* (%)			
Yes	82 (63.19)	78 (73.39)	χ^2^_4_ = 7.89 **
No	38 (28.83)	13 (11.93)	
Not answered	10 (7.98)	15 (14.68)	

** *p* < 0.01; n.s. not significant.

**Table 2 children-09-00648-t002:** Main results of hierarchical linear regressions for change in social support as a moderator of depression and stress during pregnancy.

			R^2^	F	S.E.	β	t	*p*
**Depression differences**								
	Mothers		0.146	21.40	29.44			<0.0001
		SS change				−0.227	−4.72	<0.0001
	Fathers		0.085	9.06	27.18			<0.01
		SS change				−0.168	−3.01	<0.01
**Stress differences**								
	Mothers		0.109	15.54	107.33			<0.001
		SS change				−0.371	−3.94	<0.001
	Fathers		0.105	11.74	91.41			<0.001
		SS change				−0.328	−3.42	<0.001

S.E. Standard Error; S.S. Social support.

**Table 3 children-09-00648-t003:** Conditional effect of changes in social support on differences in depression and stress.

		Depression				
		**Mean**		**S.E.**	**t**	** *p* **
		**GT-12**	**GT-32**			
Mothers	Improved SS	7.85	5.11	0.68	4.00	0.00
	No change	6.7	6.21	0.48	1.02	0.30
	Impaired SS	5.56	7.30	0.68	−2.55	0.01
Fathers	Improved SS	5.01	3.27	0.74	2.34	0.02
	No change	4.51	4.36	0.52	0.29	0.76
	Impaired SS	4.01	5.44	0.74	−1.92	0.05
		**Stress**				
		**Mean**		**S.E.**	**t**	** *p* **
		**GT-12**	**GT-32**			
Mothers	Improved SS	27.31	24.73	1.29	1.99	0.04
	No change	25.53	26.56	0.91	−1.12	0.26
	Impaired SS	23.76	28.79	1.29	−3.58	0.00
Fathers	Improved SS	24.6	20.43	1.34	3.10	0.00
	No change	23.27	22.37	0.94	0.95	0.34
	Impaired SS	21.95	24.30	1.34	−1.75	0.08

S.E. Standard Error; S.S. Social support.

**Table 4 children-09-00648-t004:** Main results of hierarchical linear regressions for relationships between social support and maternal depression/stress at GT-32 with cephalisation index and size at birth.

Mothers						
	**R^2^**	**F**	**S.E.**	**β**	**t**	** *p* **
Cephalisation Index	0.584	16.02	0.01			<0.0001
Depression at GT-32				0.066	2.61	0.01
Social Support at GT-32				0.008	1.80	0.07
Interaction				−0.001	−2.28	0.02
Cephalisation Index	0.578	15.77	0.18			<0.0001
Stress at GT-32				0.032	2.44	0.01
Social Support at GT-32				0.014	1.93	0.051
Interaction				0.000	−2.19	0.03
	**R^2^**	**F**	**S.E.**	**β**	**t**	** *p* **
Size at birth	0.389	7.25	5.26			<0.0001
Depression at GT-32				−0.892	−2.08	0.03
Social Support at GT-32				−0.140	−1.88	0.06
Interaction				0.010	1.88	0.06
Size at birth	0.375	6.99	5.28			<0.0001
Stress at GT-32				−0.380	−1.72	0.08
Social Support at GT-32				−0.200	−1.60	n.s.
Interaction				0.006	1.58	n.s.

S.E. Standard error; n.s. Non-significant.

## Data Availability

Data will be available under request to mmbellid@ugr.es or dlsantos@ugr.es.

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
