# Peer review of "Moderating Effect of Changes in Perceived Social Support during Pregnancy on the Emotional Health of Mothers and Fathers and on Baby’s Anthropometric Parameters at Birth"

_children, 2022, doi:10.3390/children9050648_

Round 1

Reviewer 1 Report

This research is interesting.

However, it may be useful to include pregnancies pathology in your model.

You may expand on the SPSS macros you used and their methodological approach.

A few typos need attention (e.g. double spaces on lines 184, 265, 298, 366, 375).

Also please review figure 1 for “Phase 2” to be within the figure limits just as “Phase 1” is. Also, describe the full names of variables in the figures to improve readability.

Redesign Figure 2, to the appropriate format.

Regarding tables, they are difficult to read; make sure are not split into different pages (table 4); tables should be redesigned so they fit the text borders. In table 4, “T” should be “t”.

Review references list to conform to their referencing style (e.g. ref. 10, 11, 14, etc)

Reviewer 2 Report

- Thank you for your effort working on this paper.

- This study is meaningful in that it was found that social support perceived by parents during pregnancy is a mediating factor for reducing parental stress and depression, and that mother's stress or depression during pregnancy is related to the baby's cephalisation index. . It is also meaningful in that it emphasizes the importance of emotional health of parents during pregnancy and guides the direction to promote emotional health. However, the following few corrections are needed to help readers understand.

- Title: It seems to be more accurate to express it as ‘baby's anthropometric parameters of the baby at birth’ rather than ‘~ on the development of the baby.’

- Line 136: Please indicate the model number, manufacturer, and country of manufacture of the measuring instrument used for anthropometric measurements.

- Line 137~138: Please indicate size → length in ‘Infant birth weight (grams), head circumference and size (centimetres) were obtained from~’.

- Table 2: It is recommended to place Depression differences and Stress differences in the same column, and to indicate Variable differences where there are Depression differences.

- Table 3: How social support was classified into improved SS and Impaired SS is not presented. Please provide classification criteria.

- Line 329~330: A more specific discussion is needed on “It is probably for this reason that no strong interaction was observed in the fathers in our study.”
